# An Immunochromatographic Assay for the Rapid and Qualitative Detection of Mercury in Rice

**DOI:** 10.3390/bios12090694

**Published:** 2022-08-28

**Authors:** Shuai Lv, Xinxin Xu, Shanshan Song, Liguang Xu, Liqiang Liu, Chuanlai Xu, Hua Kuang

**Affiliations:** 1International Joint Research Laboratory for Biointerface and Biodetection, School of Food Science and Technology, Jiangnan University, Wuxi 214122, China; 2State Key Laboratory of Food Science and Technology, Jiangnan University, Wuxi 214122, China

**Keywords:** rice sample, mercury, immunochromatographic assay, qualitatively, strip

## Abstract

Mercury is a major pollutant in food crops. In this study, we synthesized an anti-mercury monoclonal antibody (mAb; IC_50_ was 0.606 ng mL^−1^) with high sensitivity and specificity and different immunogens and coating antigens and developed an immuno-chromatographic assay (ICA) for the detection of mercury in rice. The ICA strip had a visible detection limit of 20 ng g^−1^ and a cut-off value of 500 ng g^−1^ in rice. The performance of the ICA strip was consistent with that of ICP-MS and ic-ELISA. The recoveries of mercury in rice ranged from 94.5% to 113.7% with ic-ELISA and from 93.6% to 116.45% with ICP-MS. Qualitative analysis by ICA can be obtained with the naked eye. The ICA strip is an effective and practical method for the rapid and high-throughput determination of mercury in rice.

## 1. Introduction

Mercury, one of nature’s most toxic heavy metals, causes serious harm to the environment and humans [1,2]. Mercury is the only metallic element in liquid form at room temperature, and its vapor is highly toxic [3]. Studies have reported that mercury in grains results from environmental contamination [4,5]. Once the soil and water become contaminated, the food produced in the region will have excessive levels of mercury. In excess, mercury causes damage to the kidneys, stomach, and intestines and, in severe cases, damage to the central nervous system, reproductive mutations, and DNA. The World Health Organization reported that the per capita intake of mercury should not exceed 27 µg kg^−1^. The GB 2762-2017 standard [6] issued by the China Food and Drug Administration (CFDA) in 2017 stipulates that the limit for total mercury in rice, barley, and wheat should not exceed 0.02 mg kg^−1^.

Current detection methods for heavy metals include atomic fluorescence spectrometry [7], inductively coupled plasma mass spectrometry (ICP-MS), inductively coupled plasma atomic emission spectrometry, atomic absorption spectrometry, cold atomic absorption, and heavy metal detection techniques based on antibody recognition [8].

These traditional instrumental methods, especially ICP-MS, have significant advantages, including high accuracy, strong anti-interference, and trace and ultra-trace multi-element analysis [9]. However, these methods require expensive instruments and highly trained staff and are not portable. Immunoassays can overcome the shortcomings of instrumental detection and are sensitive and have high throughput [10].

The preparation of anti-mercury monoclonal antibody (mAb) for mercury detection has been reported [11,12]. However, studies reported low sensitivity and cross-reactivity with copper [13]. Additionally, there are no reports on using immunoassay test strips in grains [14].

This study aimed to prepare a mAb with high sensitivity and low cross-reactivity by using different chelating agents for synthetic immunogens and heterologous coating antigens [15]. While developing the immuno-chromatographic assay (ICA) strip, we optimized the surface activity and antigen-antibody concentration to detect mercury in rice.

## 2. Material and Methods

### 2.1. Chemicals and Instrumentation

We obtained mercury (II), mercury (I), MeHg, cadmium (II), lead (II), copper (II), magnesium (II), arsenic (III), nickel (II), calcium (II), manganese (II), and chromium (III) at 1 mg mL^−1^ in 5% HNO_3_ from the National Analysis Center for Iron and Steel of China (Beijing, China) and isothiocyanobenzyl-EDTA (ITCBE) from Dojindo Molecular Technologics, Tnc (Osaka, Japan). J&K Scientific (Beijing, China) provided ethylene diamine tetraacetic acid (EDTA) and 6-mercaptonicotinic-acid (MNA). Keyhole limpet hemocyanin (KLH), bovine serum albumin (BSA), goat anti-mouse immunoglobulin, Freund’s complete adjuvant, Freund’s incomplete adjuvant, and gold chloride trihydrate were acquired from Sigma-Aldrich (St. Louis, MO, USA). From Gibco BRL (Paisley, UK), we obtained RPMI-1640 cell culture medium and polyethyleneglycol 1500 (PEG), and from Every Green Co. Ltd. (Hangzhou, China), we acquired fetal calf serum. All chemicals were of analytical grade or higher. We soaked the glassware in acid for more than an hour.

Thermo Fisher (Thermo Fisher Scientific, Massachusetts, USA) supplied the microplate reader, centrifuge, and ICP-MS instrument. Other equipment included an electrophoresis apparatus (HBIO, MA, USA), magnetic stirrer, and vortex (Wiggens, Beijing, China). A UV/Vis scanner (Bokin Instruments, Tsushima, Japan) was used to characterize the immunogen and antigen.

### 2.2. Synthesis of Immunogen and Coating Antigen

#### 2.2.1. Synthesis of Immunogen

Mercury, which has a molecular weight of 201 Da and no surface-active sites, cannot be directly coupled to a protein. Therefore, with some modifications, we used a bifunctional chelating agent-ITCBE based on Zhang’s [16] method. Figure 1a shows the synthesis of the immunogen. First, we dissolved 6 mg KLH (1.3 µM) in HBS, added 1.17 mg ITCBE (2.6 µM, dissolved in DMSO) dropwise, and adjusted the pH to 12 with 0.5 mol mL^−1^ NaOH. After stirring at 4 °C for 12 h, we added the mixture to 0.643 mg Hg^2+^ (3.2 µM) while ensuring that the pH was maintained at 8 with 0.1 mol mL^−1^ NaOH. To remove the excess free Hg^2+^, we filtered Hg-ITCBE-KLH three times using an ultrafiltration tube. All conjugates were adjusted to 2 mol mL^−1^ with 0.01 M HBS.

#### 2.2.2. Synthesis of Coating Conjugate

Figure 1b shows the synthesis of the coating antigens by reference to the method of Wang [17]. An amount of 0.41 mg MNA (1.76 µmol) was dissolved in 200 µL DMSO and 0.923 mg (5.29 µmol) NHS, then 1.523 mg (5.29 µmol) EDC were added. After stirring for 6 h, the activation solution was added dropwise to BSA dissolved in CB (coating buffer), and the reaction was carried out overnight. The dialysate was then dialyzed for three days to remove the excess MNA. The dialysate was solution A, and the Hg^2+^ solution was solution B. An amount of 354 µL of the B solution was added dropwise to the A solution, and the pH was controlled at around 7.4 with 0.5 M sodium hydroxide. The excess Hg^2+^ was removed by dialysis for three more days to obtain the coating antigen (Hg-MNA-BSA).

### 2.3. Preparation of Anti-Mercury mAb

All animal experiments were in strict accordance with Chinese laws and guidelines, which were approved by the Animal Ethics Committee of Jiangnan University. Female BALB/c mice (20, aged 6–8 weeks old) were assigned into two groups of 20. Referring to the previous monoclonal antibody preparation method [18,19], 20 BALB/c female mice were selected and divided into two groups. For the first immunization, a mixture of Freund’s adjuvant and immune antigen was emulsified and injected subcutaneously into BALB/c mice at three-week intervals. An indirect competitive enzyme-linked immunosorbent assay (ic-ELISA) was used to evaluate the titer and specificity of the mouse serum to mercury following each immunization. The mouse with the highest titer and specificity to mercury was chosen for cell fusion. Cell fusion was performed according to the previous literature [20]. A hybridoma fusion method was used to fuse myeloma cells, and spleen cells in a ratio of 1:10. Cells with strong signals and high specificity were selected by ic-ELISA. After three subclonations, stable antibody-producing clones were expanded and stored in liquid nitrogen. The stable subclones were injected intra-peritoneally into paraffin-primed BALB/c mice for fluid production. After a week, the fluid was extracted from the mice and purified by the caprylic acid-ammonium sulfate method to obtain mAb against mercury. After three days of dialysis in PBS, the purified mAb was stored at −20 °C [21].

### 2.4. Characterization of mAb

Sensitivity and specificity are important factors in assessing antibodies. In this research, affinity constants and maximum half inhibition were measured by indirect ELISA. We assessed the sensitivity and stability of the anti-mercury mAb by establishing a standard curve. The specificity of mAb against mercury was assessed based on the cross-reactivity (CR) [22] of mAb with Cu^2+^, MeHg, Pb^2+^, Cd^2+^, Cr^3+^, Ni^2+^, Mg^2+^, Ca^2+^, Mn^2+^, Fe^3+^.
 CR (%) = (IC_50_ of Hg^2+^/IC_50_ of other metals) × 100%.

### 2.5. Preparation of Colloidal Gold-mAb

The preparation of colloidal gold-mAb refers to the Liu’s methods [23]. Colloidal gold with a diameter of 20 nm was prepared using the sodium citrate reduction method. Briefly, 4 mL of 1% trisodium citrate solution was added to 100 mL of a 0.01% (*w*/*v*) boiling solution of chloroauric acid in a flask with vigorous stirring. The mixture was stirred continuously until the colour of the solution changed from pale yellow to burgundy. The solution was then cooled to room temperature and stored at 4 ± 1 °C to label the mAb. The colloidal gold was characterised by transmission electron microscopy (TEM). After the preparation of gold standard antibodies, the colloidal gold-mAb was stored at 4 °C for later use.

### 2.6. Preparation of the ICA Strip

As shown in Figure 2, the ICA strip contains five components: an absorbent pad, a nitrocellulose (NC) membrane, a conjugate pad, a polyvinyl chloride (PVC) plate and a sample pad [24]. The test and control lines (T- and C-lines) were generated by spraying the encapsulated antigen (0.5 mg/mL) and goat anti-mouse IgG (0.2 mg/mL) onto the NC membrane at 0.9 μL/cm, respectively, using a dispenser. The NC membranes were then dried at 37 °C for 12 h. The GNP-labelled mAb was diluted five-fold with a suspension buffer (0.02 M Tris-HCl) containing 0.1% Tween-20, 0.1% PEG, 5% sucrose, 5% alginate and 0.2% BSA. The coupling pad was then sprayed at a concentration of 3.5 μL/cm. After drying at 37 °C for 12 h, the NC membrane, binding pad, sample pad and absorbent pad were laminated and adhered to a PVC backing pad.

### 2.7. Evaluation of the ICA Strip

The optimal loading buffer solution and the concentration of test capture reagents (Hg-MNA-BSA) and mAb are first determined in the test strip evaluation and optimization process. Then, we established the standard concentration profile for mercury (0, 10, 20, 50, 100 ng mL^−^^1^), which was diluted with HBS [25]. For the evaluation, 80 µL of Hg standard solution and 40 µL of the colloidal gold-mAb were added dropwise to the ICA strip. After 15 min, the T- and C-lines will show a color reaction, and the sensitivity of the ICA will be evaluated according to the difference in color between the T- and C-lines [26]. Finally, the qualitative analysis could be performed with the naked eyes to assess the sensitivity of the ICA.

### 2.8. True Samples Characterized by ICP-MS

Four true samples (1*, 2*, 3*, 4*) were purchased from the National Research Centre for Reference Materials (NRCCRM), Chinese Academy of Metrology. The four samples were analysed simultaneously by ICP-MS and ICA strips to compare their stability and authenticity. The pre-treatment steps for ICP-MS are described in the Appendix A. As shown in Appendix A, the ICP-MS standard curve for mercury was y = 192.468 + 1022.76x with a value of R² of 0.998. As shown in Appendix A, the mercury content in the four samples was 2287.5, 507, 58.5, and 5.4 ng g^−1^, respectively, where sample number four was negative, and the rest were positive samples.

### 2.9. Sample Pretreatment

The ICA strips were used to assess the authenticity and accuracy of the actual sample [27]. The rice was powdered in a grinder, and the sample was submerged in 0.75 M nitric acid (1 g of sample in 5 mL of nitric acid solution) and vortexed continuously for 5 min. After vortexing, the sample was centrifuged at 6500 r/min for 15 min, and the supernatant was removed. The pH of the supernatant was adjusted to 7.4 with 0.75 M sodium hydroxide [28]. Five instances of dilution with HBS were used to obtain the final sample extract. Finally, the Hg standards were added to the extracts to prepare samples of different concentrations, verified by ICA test strips.

## 3. Results and Discussion

### 3.1. Antigen Characterization

We characterized the immunogen (Hg-ITCBE-KLH) and coating antigen (Hg-MNA-BSA) using UV/Vis. Figure 3a shows that ITCBE had UV absorption peaks at 265 nm, KLH had absorption peaks at 280 nm and 350 nm, and Hg-ITCBE-KLH had UV absorption in both ranges, indicating that ITCBE had successfully coupled to the carrier protein. Figure 3b shows absorption peaks at 296 nm and 342 nm for MNA, 280 nm for BSA, and 280 nm and 350 nm for MNA-BSA, indicating a successful conjugation of MNA and BSA. When bound to mercury, the UV absorption of metal ions such as Hg^2+^ is weak; the absorption peak of Hg-MNA-BSA shifted, further demonstrating the successful synthesis of the antigen.

### 3.2. Characterization of Anti-Mercury mAb

The mAb against mercury was classified by analyzing the isotype using a mouse mAb isotyping kit. Figure 4a revealed that the anti-mercury mAb belonged to the isotype IgG2b and the light chain type kappa. Following purification, we identified mAb against mercury using ic-ELISA. Figure 4b shows that the equation was y = 0.094 + (1.629−0.094)/(x/0.04)^1.539^] with a correlation coefficient (R^2^) of 0.997. The half-maximal inhibitory concentration (IC_50_) and limit of detection (LOD) of anti-mercury mAb were 0.606 ng mL^−^^1^ and 0.07 ng mL^−^^1^, respectively, with a linear range of detection of 0.302–20 ng mL^−^^1^. Therefore, anti-mercury mAb had high sensitivity and specificity [29].

The cross-reactivity (CR) of anti-mercury mAb to other metals reflects the specificity of the antibody and its tolerance to matrix interference [30]. In this study, we used two different metal chelators, ITCBE and MNA, to synthesize immunogen and coating conjugate, respectively. Antibodies against the chelators could be eliminated during the screening of the mouse sera and hybridoma cells. MNA binds to mercury much more strongly than other metal ions, which prevents cross-reactivity and increases the specificity of anti-mercury mAb. Table 1 shows that anti-mercury mAb had no cross-reactivity with other metal ions (Cu^2+^, MeHg, Pb^2+^, Cd^2+^, Cr^3+^, Ni^2+^, Mg^2+^, Ca^2+^, Mn^2+^, and Fe^3+^).

### 3.3. Performance of the ICA Strip

When tuning the performance of ICA strips, choose the appropriate concentration of colloidal gold-mAb, coating antigen, resuspension buffer and other physical parameters (ionic strength, pH). Referring to the previous laboratory optimisation, we chose 0.47% HEPES, pH = 7.4 HBS buffer in the ICA test strips to dilute the specimens. If the running buffer is not suitable, a blockage can occur, leading to color development failure, and affecting the results^27^. For optimisation, we chose basic buffer and four surfactants (5% PEG, 5% BSA, 5% Polyvinylpyrrolidone and 5% On-870) for commissioning. As shown in Figure 5a, they were evaluated by testing with 0 and 20 ng mL^−^^1^ mercury. The results showed that the basic buffer showed darker T and C color development, significant inhibition and no blockage. This indicates that the basic buffer can help the antibody and antigen react quickly to produce results. As shown in Figure 5b, the basic buffer is the optimal surfactant for the test strips. The best colour development and sensitivity when 0.5 µg mL^−1^ of antigen and 10 µg mL^−1^ of mAb were selected. Therefore, the concentrations of 0.5 mg mL^−^^1^ of antigen and 10 µg mL^−^^1^ of mAb were combined to assess the sensitivity and matrix effect of the ICA test strips.

### 3.4. Matrix Evaluation of the ICA Strip

Rice contains copper, magnesium, and macronutrients [31,32]. The presence of macronutrients interferes with the ICA strip results. To overcome this interference, a highly sensitive and specific mAb is required. Additionally, the test strip conditions need to be optimized. We chose rice as a substrate to assess the performance of the ICA strip (rice was mercury-negative by ICP-MS). According to the national standard, GB 5009.17-2014, the mercury content in grain should not exceed 20 µg kg^−^^1^. We spiked rice samples with 0, 10, 20, 50, 100, 200, and 500 ng mL^−^^1^ mercury and analyzed them using the ICA strip. The cut-off value and visual limit of detection (vLOD) were used to assess the sensitivity of the ICA strip. The vLOD refers to the minimum concentration of mercury that results in a weak T-line. The cut-off value refers to the minimum concentration of mercury that results in a colorless T-line. Figure 5c shows that at 20 ng g^−1^ mercury, the T-line became lighter than at 0 ng g^−1^; therefore, vLOD was 20 ng g^−^^1^. At 200 ng g^−1^ mercury, the color of the T-line disappeared completely. Therefore, the cut-off value was 200 ng g^−^^1^ for mercury in rice.

Therefore, despite the presence of macronutrients, the ICA strip could detect mercury in the sample with good accuracy. The ICA strip meets the national standard (0.02 mg kg^−1^) of detection for mercury in rice. The results can be visualized with the naked eye, which is convenient for the rapid determination of mercury in rice.

### 3.5. Analysis of Mercury in True Samples

Rice samples from the National Research Centre for Reference Materials (NRCCRM) of the Chinese Academy of Metrology included mercury-negative and -positive samples. The confirm the accuracy and stability of the ICA strip, standards of different concentrations of mercury (0, 10, 20, 50, 100, 200, 500 ng mL^−1^) were added to the negative samples, as shown in Figure 5c (vLOD of 20 ng mL^−1^ and a cut-off value of 500 ng mL^−1^). Additionally, we spiked mercury-negative samples with mercury and analyzed them using ic-ELISA, ICP-MS, and ICA strips. Table 2 shows the recoveries were 93.6–116.45% for ICP-MS and 94.5–113.7% for ic-ELISA.

The ICA strip qualitatively detected the concentration of mercury in rice, and the results were consistent with those obtained from ICP-MS and ic-ELISA. The ICA strip is often more reliable and less time-consuming than instrumental methods. The four rice samples were tested using mercury ICA strips. Figure 5d and Table 3 show that there were significant color differences between the T-line color of samples 1* through 3* and the blank, indicating that samples 1* through 3* were positive. Sample 3* was a weak positive sample. In contrast, there was no visible difference in the T-line color of sample 4*. The ICA strips results were consistent with the ICP-MS results.

Previously reported test strips for the detection of mercury were limited to simple matrices such as pure and river water [33,34]. In contrast, the ICA strip developed in this study can not only detect mercury in a complex matrix like rice but also has low detection costs, simple preparation methods, and low operator requirements. Additionally, it is suitable for most detection scenarios [35]. Based on this method, the identification of metal content in other grains can be further developed.

## 4. Conclusions

We synthesized different immunogens and coating antigen and screened a highly sensitive and specific mAb against mercury using ic-ELISA and hybridoma technology. We developed a GNP-based lateral-flow ICA strip assay for the rapid detection of mercury. We optimized the running buffer, colloidal gold mAb, and antigen concentration of the ICA strip of vLOD was 20 ng g^−^^1,^ and the cut-off value was 500 ng g^−1^. The mercury recovery in rice ranged from 93.6% to 116.45% (ICP-MS) and from 94.5% to 113.7% (ic-ELISA). Therefore, the ICA strip can be used to qualitatively determine mercury in the true rice, and the results are consistent with ICP-MS. Furthermore, our method can be used for large-scale mercury screening in rice and developed for other food crops.

## Figures and Tables

**Figure 1 biosensors-12-00694-f001:**
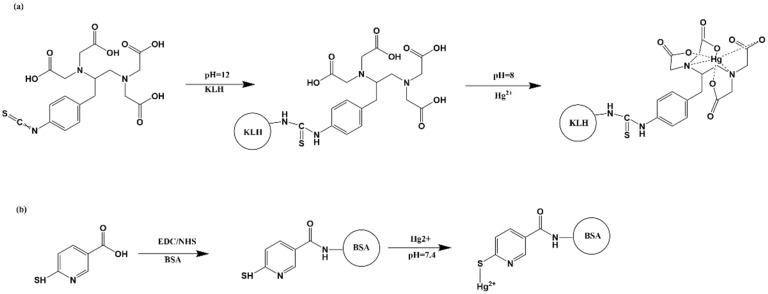
(**a**) The scheme of synthesizing immunogen (Hg-ITCBE-KLH). (**b**) The synthesis of coating antigen (Hg-MNA-BSA).

**Figure 2 biosensors-12-00694-f002:**
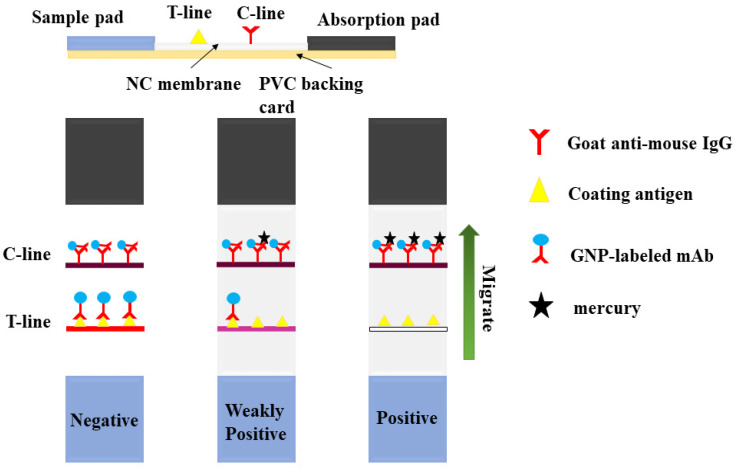
Composition of the test strip and the schematic for sample detection.

**Figure 3 biosensors-12-00694-f003:**
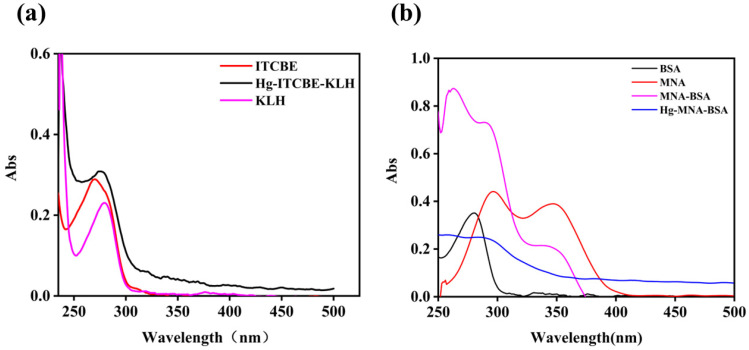
The UV–Vis spectroscopy of antigen. (**a**) Confirmation of immunogen (Hg–ITCBE–KLH). (**b**) Confirmation of coating antigen (Hg–MNA-BSA).

**Figure 4 biosensors-12-00694-f004:**
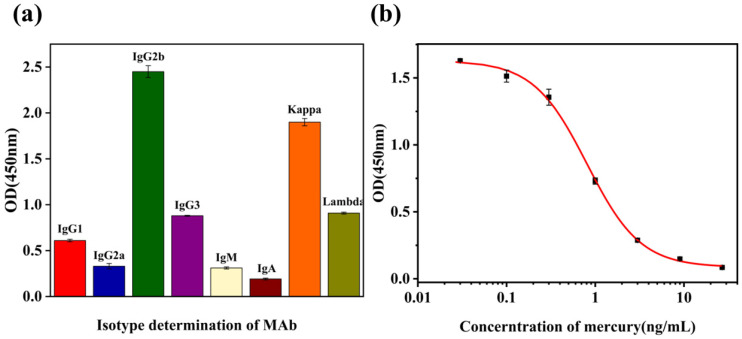
Characterization of mAb against mercury. (**a**) Isotype determination; (**b**) standard curves for mercury detection.

**Figure 5 biosensors-12-00694-f005:**
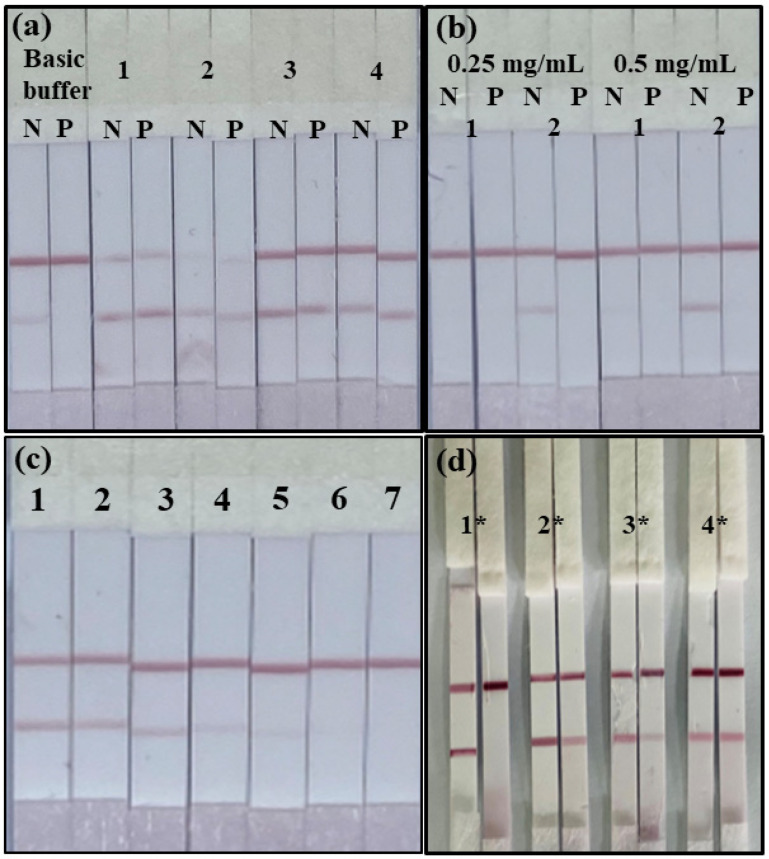
ICA strip performance and applied in the spike and true samples. (**a**) ICA strip with different running basic buffer and 1, 2, 3, 4, which represent 5% PVP, 5% On-870, 5% BSA, 5% PEG respectively (N, 0 ng mL^−^^1^; P, 20 ng mL^−^^1^ of mercury). (**b**) The optimization of the antigen and the colloidal gold-mAb concentration (1, 5 μg mL^−^^1^ mAb; 2, 10 μg mL^−^^1^; N, 0 ng mL^−^^1^; P, 20 ng mL^−^^1^ of mercury). (**c**) The numbers (1, 2, 3, 4, 5, 6, 7 represent 0, 10, 20, 50, 100, 200, 500 ng mL^−^^1^, respectively. (**d**) Sample analysis with ICA strips (*n* = 4). (1*, 2*, 3*, 4*) are four real samples of rice purchased from the National Reference Materials Centre.

**Table 1 biosensors-12-00694-t001:** The cross-reactivity of mAb against mercury with Hg^2+^, Cu^2+^, MeHg, Pb^2+^, Cd^2+^, and other metals.

Metals	IC_50_(ng·mL^−1^)	CR (%)
Hg^2+^ (Mercury II)	0.606	100
MeHg (Methyl mercury)	8.9	9.05
Hg^+^ (Mercury I)	25	3.22
Pb^2+^ (Lead)	>10,000	<0.001
Cd^2+^ (Cadmium)	>10,000	<0.001
Cr^3+^ (chromium)	>10,000	<0.001
Ni^2+^ (Nickel)	>10,000	<0.001
Mg^2+^ (Magnesium)	>10,000	<0.001
Ca^2+^ (Calcium)	>10,000	<0.001
Cu^2+^ (Copper)	>10,000	<0.001
Mn^2+^ (Manganese)	>10,000	<0.001

**Table 2 biosensors-12-00694-t002:** Analysis of mercury in rice samples by the ICA strip assay (n = 5).

Samples	Spiked Level (ng/mL)	ic-ELISA			ICP-MS			ICA Strip				
		Mean ± SD (ng/mL)	Recovery ± SD (%)	CV (%)	Mean ± SD (ng/mL)	Recovery ± SD (%)	CV (%)					
Rice	0	ND ^a^	NC ^b^	NC ^b^	ND ^a^	NC ^b^	NC ^b^	−	−	−	−	+
	10	9.86 ± 1.23	98.6 ± 4.4	4.4	9.93 ± 1.34	99.3 ± 4.66	4.6	−	−	±	+	+
	20	18.89 ± 4.23	94.5 ± 10.16	10.7	22.91 ± 4.13	114.5 ± 6.33	5.5	±	±	+	+	−
	100	112.7 ± 9.81	112.7 ± 13.59	12	93.6 ± 8.12	93.6 ± 13.12	14	+	+	+	+	±
	200	226.8 ± 13.44	113.4 ± 8.93	7.8	232.9 ± 9.87	116.45 ± 10.9	9.3	+	+	+	+	+

Notes: ^a^, ND, not detectable; ^b^, NC, not calculated. −, negative: the concentration of Hg^2+^ was <5 ng mL^−1^; ±, weakly positive: the concentration of Hg^2+^ was 5–20 ng mL^−1^ in rice samples; +, positive: the concentration of Hg^2+^ was ≥20 ng mL^−1^ in rice sample.

**Table 3 biosensors-12-00694-t003:** Analysis of mercury in true samples by the ICA strip assay (n = 4).

Sample	ICP-MS	Visual			
1* Rice	2287.5 ± 50.23	+ ^a^	+ ^a^	+ ^a^	+ ^a^
2* Rice	507 ± 17.66	± ^b^	+ ^a^	+ ^a^	+ ^a^
3* Rice	58.5 ± 8.17	± ^b^	± ^b^	± ^b^	+ ^a^
4* Rice	5.4 ± 0.83	− ^c^	− ^c^	− ^c^	± ^b^

Notes: ^a^, positive; ^b^, weak positive; ^c^, not detectable.

## Data Availability

Not applicable.

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
