# Peer review of "An Immunochromatographic Assay for the Rapid and Qualitative Detection of Mercury in Rice"

_biosensors, 2022, doi:10.3390/bios12090694_

Round 1

Reviewer 1 Report

In this manuscript, GNP-based lateral-flow ICA strip assay was developed for the rapid detection of mercury. In this developed ICA chip, immunogens and coating antigens were utilized for the specific capture of mercury ions. The authors claimed that the ICA strip had a visible detection limit of 20 ng g-1 and a cut-off value of 500 ng g-1 in rice. Even though I agreed this sensor is sensitive compared to the ICP-MS method, I could not understand the competitive assay is better than the conventional ICA method. In a typical ICA, antigen against a target antigen (in this study, anti-mercury) is fixed on the strip. This method looks simple and more accurate method than the suggested method. Please provide a reasonable description or comparison data for this issue.    

Reviewer 2 Report

In the present study, Lv et al. synthesize effective anti-mercury mAB and use it to develop a ICA strip to qualitatively check for Hg2+ contamination in rice. The manuscript is very well written and presents the data clearly. The results are supported by appropriate and comprehensive experiments. The experiments presented are clear and all controls are done properly.  In my opinion, the manuscript is appropriate for publication in Biosensors as is or with minor revisions.

Minor Revisions:

1.     One things author can perhaps test is the effect of various physical parameters (Ionic strength, pH etc.) on the performance of the ICA strip

2.     In the figure 3b, where authors show coupling of MNA-BSA to Hg, there is a disappearance of peak around 350nm in the MNA-BSA-Hg adduct. Can the authors comment on that?

Round 2

Reviewer 1 Report

The authors provide an appropriate response. It could be accepted in this journal.